# Silica-Filled Polyacrylonitrile Solutions: Rheology, Morphology, Coagulation, and Fiber Spinning

**DOI:** 10.3390/polym14214548

**Published:** 2022-10-27

**Authors:** Lydia A. Varfolomeeva, Ivan Y. Skvortsov, Mikhail S. Kuzin, Valery G. Kulichikhin

**Affiliations:** A.V. Topchiev Institute of Petrochemical Synthesis of Russian Academy of Sciences, Leninsky Av. 29, 119991 Moscow, Russia

**Keywords:** polyacrylonitrile, silica, rheology, fiber spinning, fiber morphology, suspensions, structured solutions

## Abstract

The fumed silica influence on the morphology, coagulation processes, and rheological properties of suspensions in dimethyl sulfoxide (DMSO) and polyacrylonitrile (PAN)–DMSO solutions has been studied for the production of composite films and fibers. It has been shown that silica–DMSO concentrated suspensions (24 wt%) form a weak gel with a yield point of about 200 Pa. At concentrations of ~5 wt% and above the dispersions, depending on the shear stress, are pseudoplastic or dilatant liquids. It has been found that the silica addition method into a PAN solution has a significant impact on the aggregates dispersibility and the rheological behavior of the obtained systems. A thixotropy appearance and a sharp increase in the relaxation time were observed for PAN solutions at a SiO_2_ content of more than 3−5 wt%, which indicates the formation of structures with a gel-like rheological behavior. Upon reaching the critical stress their destruction takes place and the system starts to behave like a viscoelastic liquid. Two spinning methods have been used for preparing fibers: standard wet and mechanotropic. By the mechanotropic method it is possible to achieve a higher draw ratio at spinning and to obtain fibers with better mechanical properties. It is possible to spin fibers from PAN solutions containing up to 15 wt% of silica per polymer with a tensile strength up to 600 MPa.

## 1. Introduction

In addition to the traditional application of polyacrylonitrile PAN fibers for wool-like fabrics, special purpose products are used in the textile industry for protective clothing manufacture (chemical protection, special military uniforms, self-cleaning materials, medical products, sportswear) [1,2]; membranes’ production for the separation of oil-water emulsions in the oil industry [3]; separators in lithium-ion batteries [4,5]; and, of course, as carbon fiber precursors [6,7]. To best fulfil the requirements for such materials, additional modification to impart increased hydrophobicity and variation in ionic conductivity is needed. This can be achieved by introducing various silica species into the composite materials.

There are various ways to obtain silica-filled composites, and each of them has advantages and disadvantages. Many methods are described for introducing silicon-containing items into PAN via sols obtained as a result of the hydrolytic polycondensation of tetraethoxysilane (TEOS) in PAN solutions [8]; the synthesis of PAN in a silica-containing sol [9]; the impregnation of PAN fibers with organosilicon substances followed by the carbonization of the obtained precursors [10]; the synthesis of SiO_2_ particles and oligosiloxanes from TEOS [11,12] and vinyltriethoxysilane [13] in PAN solutions with the subsequent formation of silicon carbide particles in a carbonized matrix [12,13]; an introduction of silicon nanoparticles into PAN dopes for preparing the carbon-SiC fibers [14,15]; and coating carbon fibers by SiO_2_ dispersion followed by heat treatment and obtaining a SiC shell on carbon fibers [16,17,18,19]. Several publications are devoted to the production of various non-woven materials by electrospinning from suspensions based on PAN solutions containing silica nanoparticles [20,21,22,23].

Silica is a commercially available product. With a relatively small size of primary particles of 6–40 nm, which are determined by the synthesis conditions, silica has a specific surface area of up to ~400 m^2^/g but tends to form aggregates [24]. In aqueous media, the interaction between the particles can be controlled by varying the ionic strength or pH (addition of salts) to reach at least the kinetic stability of the silica particle dispersions. These phenomena were studied in several publications [25,26,27], and mechanisms describing the conditions of stability/instability were proposed in [28], where the effect of medium acidity on stability, the rheological properties of aqueous suspensions, and the sizes of aggregates were studied.

The rheological properties of filled silica low molecular weight liquids have been developed in [21,28,29,30]. The authors [29] have analyzed the rheology of hydrophilic silica (A200) dispersions in a wide range of organic solvents. It has been shown that silica particles form stable low-viscosity sols in lower N-alkanols and oligomeric glycols by hydrogen bonds formed between silanols on the particles’ surface and a solvent with the formation of a solvation layer that stabilizes the particles. In other liquids such as higher n-alkanols, cyclic carbonates, and end-capped oligoethers, silica dispersion flocculates into colloidal gels via the formation of hydrogen bonds between the particles’ surface.

From the point of view of silica introduced into commercial PAN solutions, the compatibility of this additive with aprotic solvents is of practical interest. Data on silica–DMSO suspensions stability were discussed in [28], where the aggregation processes of unmodified and surface-modified silica by exchanging the silanol groups with Si–O Li+ followed by treatment with 1,3-propanesultone in DMSO were studied by dynamic light scattering. It has been shown that surface modification affects the degree of sol aggregation, and high silica concentrations in DMSO lead to the formation of a very weak gel which can be broken by sonication. Particle modification significantly reduces the number of Si-OH silanol groups on the surface and hence hydrogen bonding sites, which is a possible reason for the slower aggregation in these dispersions.

Polymer addition into a silica suspension leads to a significant change in its properties. The affinity of a polymer either for a solvent or silica can contribute both to the stabilization of individual particles or the increase in their aggregation [30,31]. Changing the polymer concentration with the same silica content can change the suspension’s rheological properties radically. Thus, for silica suspensions in high molecular weight polyethylene oxide solutions [30], a completely different rheological behavior is realized: rheopexy, thixotropy, non-Newtonian, or viscoplastic behavior up to gel formation at definite polymer concentration.

There are few publications about the silica addition effect on the properties of PAN solutions and hybrid fibers. It was shown in [32] that the silica addition changes the polymer crystallization kinetics and significantly increases the characteristic relaxation times at low frequencies. At the same time, there are several papers about obtaining functional materials with silica and studying their properties. For example, it was shown in [2] that the deposition of hydrophobic SiO_2_ particles on PAN fibers makes it possible to modify their surface, such as by attaching hydrophobic properties to the material while maintaining vapor and air permeability. The obtained fabrics had good abrasive resistance and retained their properties in acidic and alkaline media.

Similar results were obtained for composite non-woven membranes obtained by the electrospinning of PAN solution in dimethylacetamide (DMAc) and polyurethane (PU) solution in DMAc, containing perfluorodecyltriethoxysilane-modified silica from two spinnerets [1]. The composites possessed super hydrophobicity due to modified PU fibers and retained such properties as vapor permeability and high mechanical characteristics that were inherent to the PAN membranes.

PAN fibers doped with SiO_2_ are promising candidates for the fabrication of high-performance membranes in lithium-ion batteries [4]. The fibers in this study were obtained by centrifugal spinning from a 17 wt% solution of PAN in dimethylformamide (DMF) with a silica content of up to 12 wt%. Their addition led to a decrease in the fiber diameter from 1.4 μm to 0.8 μm at a content of 12 wt% silica, which may be attributed to the repulsive force between SiO_2_ particles that minimizes the entanglement of polymer chains. It is shown that the resulting membranes have a significantly higher porosity compared to membranes from polypropylene fibers. Such unwoven membranes outperform traditional polyolefin-based membranes due to better wetting with polar liquid electrolytes, and the addition of SiO_2_ improves electrochemical characteristics, such as ionic conductivity. Hybrid membranes based on PAN fibers filled with silica and graphene oxide for separating oil and water emulsions were described in [3]. The additives had both a positive effect on the separation and mechanical properties of the membranes. The authors noted the formation of inhomogeneities in the fibers when introducing SiO_2_ into the spinning solution. Judging by the morphology of the fibers, the addition of silica leads to the formation of the surface roughness and its thinning–thickening, which may not always be useful in the case of the manufacture of textile fabrics.

The morphology of silica-filled fibers was studied in detail in [33], where the preparation of PAN fibers containing up to 6 wt% of silica added to the PAN:DMF spinning solution, and an analysis of changes in the surface of the fibers with an increase in the additive concentration was performed. Interestingly, there exists little information in the available publications on the methodology of adding silica that is prone to the formation of stable aggregates into PAN solutions, despite the importance of this procedure not only from the viewpoint of the final properties of silica-containing materials, but also from the position of the finer variation of the rheological properties of the dopes.

Based on the above-mentioned information, we plan to touch on three important sections in this paper. In the first one, the rheological behavior of silica–DMSO dispersions is studied in a wide concentration range, and the dimensions of silica aggregates are estimated. In the second section, the effect of the silica presence in PAN solution in DMSO on the rheological properties, morphology, and interaction with a coagulant are tested. Particular attention is paid to the influence of the method for the silica addition into the polymer solution on the enumerated above characteristics. Finally, the appropriate method of composite fibers spinning is chosen and tested, and their morphology, structure, and mechanical properties are studied.

## 2. Materials and Methods

### 2.1. Materials

Terpolymer PAN 316020 (93.9% acrylonitrile, 5.8% methyl acrylate, 0.3% methyl sulfonate, M_w_ 85 kg/mol, polydispersity index 2.1) was supplied by Good Fellow, Huntingdon, UK. DMSO (99.5%, produced by Ecos-1, Moscow, Russia) was used as a solvent for PAN and a dispersion medium for silica. Hydrophilic fumed silica was used without surface modification (Aerosil A-380) (average size of individual particles ~7 nm, specific surface according to BET 380 ± 30 m^2^/g produced by Evonik, Essen, Germany).

PAN and silica were dried on a rotary evaporator under a vacuum of 0.05 bar at 100 °C for 4 h to constant weight before the experiments. DMSO was additionally dried over A4 molecular sieves manufactured by Sigma Aldrich (Darmstadt, Germany).

#### 2.1.1. Silica–DMSO Dispersions

Silica–DMSO dispersions up to 24.4 wt% of solid were prepared using intensive mechanical mixing with an anchor J-shaped rotor for one hour at a stirrer speed of 1000 rpm, followed by dispersion in an Elmasonic S 15 ultrasonic bath (Elma Schmidbauer GmbH, Singen, Germany) at 20 °C for one hour, followed by treatment with an ultrasonic immersion probe UZDN-A (Lviv, USSR) with an exposure frequency of 22 kHz and a power of 130 W for 0, 20, or 40 min. To prevent overheating, the dispersions were contained in a water-cooled bath with a temperature of ~15 °C during sonication. Then, the dispersions with 0.05 wt% silica concentration were made for particle size measurements by dynamic light scattering. All samples were sealed by a silicon cap to exclude watering by air humidity.

#### 2.1.2. Silica-Filled PAN–DMSO Solutions Preparation

PAN solutions with a silica addition were prepared in different ways (S1–S6) using varying dispersion methods (a–d), and different procedures were also employed to add PAN into the blend. All prepared compositions are presented in Table 1.

Systems S1–S4 were prepared by the dilution of concentrated 22 wt% SiO_2_–DMSO dispersion (made according to the method indicated in Section 3.1); for preparing the S5 system silica and PAN powders were mixed first, and then the solvent was added; in the S6 system, a prepared PAN–DMSO polymer solution (33 wt%) was added to the concentrated silica–DMSO dispersion (made according to Section 3.1).

For the model experiments, the compositions were prepared with a silica content of 3 wt% relative to the mass of the solution, and the content of PAN in DMSO was 20 wt%.

To study the influence of the silica concentration on the morphology and rheological properties of the filled solutions, the 20 wt% PAN solutions containing 0.3 to 5 wt% of silica (to the solution weight) prepared by the S4 method were studied.

To obtain fibers containing silica, three-component systems’ PAN–DMSO–silica were prepared (method S4) with a polymer content of 33 wt%, and 0.3, 1, and 3 wt% of silica at solution weight, or 0.9, 3.0, and 9.0 wt% in the fibers obtained by the mechanotropic method, and 3 wt% of silica at solution or 15 wt% in fibers obtained by the wet method. Mixing was carried out through the so-called “opposing mixer” (Figure 1) developed in the laboratory, which homogenizes the solution by periodically passing in direct and reverse directions (at least 1000 cycles) through a multi-orifice capillary with a 1 mm diameter each.

#### 2.1.3. Film Preparation

Films with a thickness of 200 μm were obtained by the solvent casting method pouring systems S1–S6 at a constant polymer concentration of 20 wt% in DMSO and a silica content of 3 wt% relative to the mass of the entire solution. The films were dried for 12 h to a constant weight with a gradual increase in temperature from 40 to 70 °C.

### 2.2. Methods

#### 2.2.1. Rotational Rheometry

The rheological behavior of the obtained systems was studied on a rotational rheometer MCR301 (Anton Paar, Graz, Austria) in continuous and oscillatory modes of shear deformation, using a plate–plate geometry with a diameter of 25 mm and a gap between the plates of 0.5 mm. The flow curves were obtained in the shear rate range of 10^–3^–10^3^ s^–1^ in the regimes of both stepwise increases until the sample began to flow out of the gap at high rates and decrease in the shear rate; the frequency dependences of the storage and loss moduli were measured in the linear region of viscoelasticity at frequencies from 0.628 to 628 rad/s. All measurements are performed at 25 °C.

#### 2.2.2. Modeling of the Coagulation Process

To evaluate the effect of the introduced silica on the coagulation process at wet and dry-wet jet spinning, the coagulation was simulated on drops of the silica-filled PAN solutions placed into a gap of constant thickness between the slide and cover slip. A coagulant based on water and DMSO with a ratio of components of 15:85 was used. The samples were prepared in a glove box with a relative humidity of less than 1% to exclude the effect of air humidity on the coagulation process. The method is described in detail in [34].

#### 2.2.3. Transmission Optical Microscopy

The surface morphology of PAN films containing silica, as well as the droplets coagulation of filled compositions, was studied using a Biomed 6 PO optical polarizing microscope (Biomed, Moscow, Russia). Images were captured using a ToupTek XFCAM1080PHD camera (ToupTek, Hangzhou, China).

#### 2.2.4. Transmission Electron Microscopy

A transmission electron microscope LEO 912 ab Omega (LEO Carl Zeiss SMT Ltd., Jena, Germany) at an accelerating voltage of 100 kV was used. To study the morphology of fiber cross sections, the samples were prepared as follows. A bundle of fibers was poured with an epoxy resin to obtain a microplastic in the form of a cylinder with a diameter of ~500 μm. Using the Ultracut R Ultramicrotome (Leica Microsystems, Wetzlar, Germany), slices were cut with a thickness of 100 nm perpendicular to the fiber long axis. The resulting cut was fixed on a copper mesh substrate.

#### 2.2.5. Dynamic Light Scattering

Measurement of the size distribution of silica particles was performed by dynamic light scattering in a Zetasizer Nano-ZS (Malvern Panalytical, Malvern, United Kingdom) analyzer at a scattering angle of 173°. The data are averaged over 10 separate measurements for each sample. The experiments were carried out for dilute suspensions with a concentration of 0.05 wt% to ensure the Brownian motion of the particles under study.

#### 2.2.6. Fiber Spinning and Characterization

The fibers were spun by two methods. The first method was the so-called mechanotropic method, which was developed earlier in our laboratory [35]. According to this method, the phase separation of the solution jet proceeds as a result of strong extension. The spinning procedure was performed on a laboratory stand (Figure 2a) with a monofilament spinneret with a diameter of 500 μm. The as-spun fiber that was formed after jet extension was subjected to three stages of treatment, including orientation stretching, drying at 80 °C, followed by thermal stretching at 110 °C.

The second method is the traditional wet spinning procedure (Figure 2b), using a coagulation bath sequence with a gradual increase in water content in the coagulants. The first bath consisted of 85 wt% DMSO and 15 wt% water; the second and third bath had a composition of DMSO:water that was equal to 70:30 and 50:50, respectively. To measure the fiber diameter, the Biomed 6PO (Biomed, Moscow, Russia) optical microscope coupled with a Touptek XFCAM1080PHD (Touptek, Hangzhou, China) camera with a magnification of 60×, which has an accuracy of ±0.3 μm, was used. For every sample, at least 10 filaments were examined. The inhomogeneity was characterized by the ratio between the maximum and minimum of the fibers’ diameters.

The mechanical properties of the fibers were measured using an Instron 1122 Tensile (Instron, Norwood, Massachusetts, United States) machine with a basic filament length of 10 mm. All measurements were performed at 23 ± 2 °C and an extension speed of 10 mm/min. The reported results were averaged for at least 10 tests.

## 3. Results and Discussion

### 3.1. Silica–DMSO Suspension

The dynamic light scattering method was used to determine particle sizes in a 0.05 wt% silica–DMSO suspension after homogenization in a rotor-stator homogenizer and subsequent sonication for 0, 20, and 40 min.

The light-scattered intensity of dispersion (Figure 3a) is proportional to particle radius to the 6th power; therefore, for particles of complex shape or systems with a wide size distribution, the signal from large particles significantly exceeds the signal from small particles. That is why the most obvious is the distribution over the volume occupied by the particles (Figure 3b).

Figure 3a shows an almost monomodal distribution, and the main signal is created by large particles with a size of ~120 nm, but when observing a more relevant coordinate (the volume fraction), the contribution of particles with sizes <100 nm becomes visible. After 20 min of sonication, the ratio between the 120 nm and 50 nm fractions is 4:1, and after 40 min it becomes 1:1 (Figure 3b). The further reduction in the size of the aggregates no longer occurs, therefore, a longer treatment time is not efficient at sonicating this power. The more powerful sonication may contribute to the DMSO destruction as a result of local overheating, which means a limitation of completely breaking the SiO_2_ aggregates in DMSO.

The results obtained are in good agreement with the well-known fact that the dispersion efficiency decreases with a decreasing particle size: the smaller the size, the higher the increase in the interfaces and the more time and energy required for further aggregates grinding [36].

It should be noted that the obtained suspensions are quite stable: the size distribution did not change during a week; therefore, the sonication for 40 min was chosen for further studies.

### 3.2. Silica–DMSO Rheology

The rheological behavior of silica dispersions in DMSO was studied by rotational rheometry in a steady-state (Figure 4a) and oscillatory (Figure 4b) shear modes over a wide concentration range.

At silica content in a dispersion of 5 wt%, the viscosity increases approximately 10 times (Figure 4a), and the system starts to exhibit pseudoplastic properties at stresses above 1 Pa. With the increase in silica concentration, signs of structure formation appear, and in the region of the content of more than 5 wt%, dualism in the behavior of the dispersion becomes noticeable: with the growth in the shear rate, the viscosity at first decreases, and then increases. It is possible to explain such dualism by destroying the initial structure to a certain limit, and this domain of shear rates is interpreted as pseudoplastic (or shear thinning) behavior. However, with a further increase in the shear rate, the resulting fine aggregates are assembled under the action of deformation into larger ones, and this domain should be considered as a dilatant (or shear thickening) behavior. After critical stresses, the onset of instability is observed. Similar behavior was noted for silica suspensions in polyethylene glycol over a wide range of concentrations [37].

Highly silica-filled dispersions (above 20 wt%) have a clear yield point and elastic and loss moduli cease to depend on the frequency. In addition, the elastic modulus becomes significantly higher than the loss modulus (Figure 4b), i.e., the suspension transforms into a weak gel.

### 3.3. Influence of Suspensions Preparation Method on Their Properties

The sequence of PAN powder addition and dispersion methods used into the silica–DMSO suspension has a significant effect on the SiO_2_ distribution. To select the optimal method, model dispersions with 3 wt% of silica in a 20 wt% PAN solution were prepared (samples S1–S6) and their rheological behavior and morphology were studied. Due to the similar refraction indices of silica (n20/D 1.4734) and DMSO (n20/D 1.4745), the study of suspensions’ morphology by optical microscopy is not informative. Therefore, to estimate the volume distribution and aggregate sizes, films were cast from the polymer containing mixtures (refraction index of PAN is 1.514), which after drying, were studied by optical microscopy.

The character of SiO_2_ distribution in polymers has a strong influence on their rheological properties. This was a reason to first consider the flow curves of silica-filled PAN solutions, as shown in Figure 5. Depending on the experimental method of combining the components, the suspensions’ properties differ significantly. For comparison, the corresponding dependence for a 20 wt% PAN–DMSO solution is also shown.

In the case of a standard test with a stepwise increase in shear rate (Figure 5a), the results are not clear, and extracting the principal features of the rheological behavior of different systems is impossible. In the case of experiments starting from high shear rates (Figure 5b), with a gradual decrease in the rate to the minimum values, the behavior of the systems becomes clearer. All suspensions are liquids showing the viscoplastic behavior, and in terms of the yield stress values they differ in the following order: S2 > S1 > S3 > S4 > S5 > S6. In the area of shear thinning behavior, suspensions behave approximately in the same manner, although it can be noted that system S4 has the highest viscosity in this domain, system S5 exhibits weak dilatant properties, and system S6 is the least structured. The yield point observed in the S6 system in the experimental set with an increasing shear rate (Figure 5a) may not be due to the internal properties of the system, but the presence of large aggregates with dimensions exceeding the gap between plates.

These differences can be explained by the long relaxation times of the systems and, accordingly, the impossibility of achieving equilibrium values of the measured stress at a given shear rate in the real time range of the experiment.

Let us consider in more detail the dynamic properties of the obtained systems depending on the preparation method. Figure 6 shows the frequency dependences of the elastic and loss moduli for the systems under study. For comparison, in all the plots of Figure 6, the corresponding dependences for a 20 wt% PAN:DMSO solution are presented by lines without dots. These dependences were compared with the morphology of PAN films obtained from the corresponding suspensions (Figure 7).

When analyzing the frequency dependences of the moduli, the following considerations were taken into account. First, a low difference in the values of the storage and loss moduli in the low-frequency region. Second, the low values of the slope of the corresponding dependences in this area compared with the prediction of the linear viscoelasticity theory. From this point of view, the compositions obtained by methods S1−S3 are highly structured systems, since the difference in the values of the loss and elasticity moduli is small and the slopes of the linear sections of dependences in the terminal zone are close to each other.

The reasonable differences in the values of moduli are observed for suspensions obtained by methods S4 and S5, and for them, there exists a prominent difference in the slopes of the frequency dependences of the elastic and loss moduli. The behavior of the S6 system is drastically different, since at low frequencies the elastic modulus exceeds the loss modulus that is typical for strong-structured, gel-like systems.

The data obtained indicate the need for the integrated approach and consideration of a combination of volumetric (rheological) and morphological (surface) methods. According to rheological data, for the best correspondence of the moduli location and its tempo of increase with frequency, system S4 is preferable. As is seen from the morphological images of the compositions under investigation, in this composition the most uniform particle distribution is observed. Based on this approach, it was concluded that the method for preparing S4 composition is optimal, therefore, the samples studied further were prepared according to this method.

The above-mentioned part of this research is more preparative than scientific, but it is extremely important for subsequent parts because the character of silica particles distribution plays a key role in further analysis and the interpretation of results.

### 3.4. Rheology of PAN–Silica–DMSO Systems

The following section concerns the filled solutions with a much higher content of PAN as the preliminary step to a search for spinning solutions. The most crucial problem consists of reasonable silica content in dopes allowing stable spinning. One of the limitations is the unusual rheological behavior of the filled PAN solutions in DMSO.

Flow curves for the 20 wt% PAN solution containing various silica concentrations are shown in Figure 8.

Silica at content up to 1 wt% does not affect the solutions’ rheology. With a further increase in the concentration, the suspensions become structured and exhibit yield behavior with a yield stress of ~1 Pa for the system with 3 wt% silica and ~10 Pa for the 5 wt% suspensions. As is seen in the cycles’ increase and decrease in shear rate, the silica-filled PAN solutions are thixotropic, forming a structure that is easily broken with an increase in shear stress, but restored under a change of strain intensity in the opposite direction.

The frequency dependences show that the effect of silica begins to be significant from 3%, resulting in an increase in the elastic and loss moduli with a corresponding decrease in the slope values, which indicates the structuring of the system. For 5 wt% silica content, the slope values in the terminal zone become close each to other (Table 2).

The concentration dependence of viscosity at different shear stresses allows us to determine a weak structure formation region, which is formed by silica aggregates (Figure 9). At low shear stress, starting from a silica content of 3 wt%, the viscosity growth intensity increases, while at more high stresses it remains on a level of moderate change. The existence of critical stress (50 Pa) leading to destroying the physical structure opens up the possibility of obtaining highly filled solutions and, consequently, fibers filled with silica.

Until now, the nature of the interaction between PAN solution and silica has not been discussed, although it is important for understanding the rheological properties of multicomponent compositions as a whole and suspensions partially. The change in the relaxation properties of the polymer solution in the presence of SiO_2_ particles will be the most indicative. Previously, this approach was applied to another silicon-containing component in the PAN solution, and either a decrease in the intrinsic relaxation times due to the dilution of the system with tetraethoxysilane (TEOS) (in a solubility range) [11], or, conversely, their growth in a narrow concentration range due to the formation of emulsions with TEOS or polydimethylsiloxanes were observed [38]. The effect of solid nanoparticles was studied in [39], where an increase in the relative relaxation times due to the interaction of particles with the polymer and the limitation of the free volume in the polyethylene melts was observed.

Relaxation times were calculated using Equation (1) [40]:(1)$$\lambda =\frac{G^{\prime }}{\left|\eta^{*}\right|\cdot \omega^{2}},\;$$
where |*η**| is the complex viscosity, *ω* is the angular frequency, and *G*’ is the storage modulus. The relative relaxation time (Δ*λ*) was calculated as the ratio of the corresponding times of the mixture system (λ_PAN+DMSO+silica_) and the initial solution (λ_PAN+DMSO_) determined at the same frequency.

Figure 10 shows the dependences of Δ*λ* at different frequencies for a 20 wt% PAN solution on the silica concentration. It is seen in these coordinates that the relaxation behavior of the systems practically does not change up to a particles content of 1 wt%, and a drastic transformation of properties at a silica concentration above 3 wt% is observed.

For solid particles, the infinite cluster formation concentration (*φ_m_*) decreases with decreasing particle size and can be estimated from Equation [41]:(2)$$\frac{1}{\varphi_{m}}=\frac{0.06}{\sqrt[{3}]{d_{eq}}}+1,\;$$
where *d_eq_* is the equivalent particle diameter.

Based on the size of the individual particles of 20 nm, the calculated concentration of the percolation threshold for silica is ~4%, (or 6.7% for 80 nm aggregates), which corresponds to the strong increase in the relaxation time and the appearance of thixotropy for the filled systems.

### 3.5. Morphology of the PAN–Silica–DMSO Films

A film was cast from a composition containing 3 wt% silica per solution (15 wt% per polymer), and the morphology of its section was visualized using the TEM method (Figure 11).

One can see in a thin 100 nm section loose aggregates formation, which are possibly filled with polymer and occupy up to 50–60% of the film volume. It approximately corresponds to the calculated maximum filling degree. Individual aggregates of about 10–20 nm in size form fractal agglomerates, which looks like a continuous branched network.

### 3.6. Coagulation

The process of obtaining fibers from polymer solutions proceeds via the phase separation of solution, either as a result of mass transfer (interdiffusion of the solvent and coagulant) in the case of wet spinning [42], or as a result of the strong stretching of the jet (mechanotropic spinning) [43]. Wet spinning imitation is conveniently carried out on a drop of solution (analogous to the fiber cross-section) surrounded by a coagulant. The presence of a dispersed filler which gives solutions thixotropic properties introduces specifics into the coagulation process. Images of the coagulated droplets of the initial and filled solutions are shown in Figure 12.

In a wet spinning process, the formation of a skin-core morphology is inevitable, and it appears when a drop of PAN solution is coagulated even with a soft coagulant (mixture of DMSO:water = 85:15). The same situation with forming a surface layer, different from the core, is also observed for silica-contained compositions. The front of solvent movement is more intensive compared with the coagulant front, but the cluster structure of silica in the filled solution slows down the interdiffusion, which makes coagulation more uniform and reduces the dimensions of finger-like defects [11,42], as was seen in rheologically similar systems with a yield point. The presence of silica aggregates in the skin can enhance the strength of the surface layer of the fibers and prevent the propagation of coagulation defects (indicated by the arrow in Figure 12) during spinning.

### 3.7. Fiber Spinning

To assess the influence of the spinning method on the mechanical properties of silica-contained fibers, two series of comparative spinning procedures by wet and mechanotropic methods were carried out. The silica concentrations relative to the polymer were 0.9, 3.0, and 9.0 wt% for the mechanotropic method and 15 wt% for wet spinning.

### 3.8. Mechanotropic Spinning

To assess the possibility of realizing high stretching ratios, the mechanotropic method [35] was chosen, allowing us to use the large spinneret holes that prevent flow instability, and to apply high stretching ratios to reach the required thickness of the fibers.

Mechanotropic spinning proceeds without any coagulant, but for its realization, spinning solutions with certain viscoelastic characteristics are required. In particular, the viscosity must be high enough to prevent sagging and the viscous breakage of the liquid filament, and the elastic response must be slightly greater than the viscous one to permit the orientation stretching. For this purpose, 33 wt% solutions of PAN with silica were prepared. The flow curves and frequency dependences of the dynamic moduli of the suspensions at 70 °C are presented in Figure 13.

There is no yield point for all systems. The values of dynamic moduli increase with an increase in the content of the silica, and simultaneously, the degree of the structuring of the systems increases. However, in the terminal zone, the loss modulus always exceeds the elastic one and these systems are not gels.

A series of fibers were spun from the solutions by the mechanotropic method. It has been found that the presence of silica has a negligible effect on spinnability and does not affect the draw ratio, allowing well-drawn filled fibers similar to the neat ones to be obtained. The mechanical characteristics of the spun fibers are presented in Table 3, and their images by optical microscopy are shown in Figure 14.

As is seen, a mechanotropic method allows for obtaining high-strength PAN fibers. The silica addition leads to some increase in the values of the achievable stretching (spinbond hood) but leads to a 20 wt% decrease in fiber strength, while such properties are still good enough for any textile production or their subsequent carbonization. All fibers are transparent, without visible defects, with a smooth surface and equal thickness, which indicates the correct spinning conditions chosen.

Of particular interest is the question of the silica distribution into the spun fibers. TEM images of the fiber cross-sections taken with different magnifications are shown in Figure 15.

It is seen that the silica aggregates are evenly distributed throughout the volume of the fiber. Inclusions with a size from 50 to 150 nm are at a distance of 150–500 nm from each from other. Opposite the previously observed phenomenon, where TEOS drops migrate during spinning and concentrate near the fiber surface, the silica aggregates distribute more or less uniformly into the fiber cross-section. This may be explained by the different mobility of liquid drops and solid particles. The other explanation consists of the fast solidification of the stretched jet, resulting in the fixation of the particles location.

### 3.9. Wet Spinning

Table 4 presents the mechanical properties of fibers obtained from a 20 wt% PAN solution and the same solution with the addition of 3 wt% silica (to solution weight).

The addition of even a high concentration of silica does not lead to a change in the spinning conditions and the stretching modes during orientation, and as a result, does not affect the fiber’s mechanical characteristics. However, it significantly affects its diameter, which is probably due to the high content of the introduced component per polymer (15 wt%). The images of the resulting fibers are shown in Figure 16.

The fibers obtained are uniform in thickness, clear, and have no visible defects. The soft coagulant allows the preparation of defect-free fibers. However, the maximum achievable draw ratio for wet spinning is significantly lower than in the case of mechanotropic spinning. This phenomenon may be associated with a different mechanism of polymer solution jet solidification: in the case of wet spinning, coagulation starts on the fiber surface, resulting in a stiff shell and softer core. During initial draw in the coagulation bath of such as-spun fiber, the initial stretching of the shell occurs; meanwhile, the central part remains liquid up to the complete coagulation of the jet. The subsequent stretching of the fully coagulated fiber is limited by the already drawn shell, which can form cracks, resulting in a fiber break. As a result, such heterogeneity in the cross-section orientation of the fiber affects the mechanical characteristics, reducing the strength.

## 4. Conclusions

Silica A380 forms stable aggregates in DMSO with a size of ~100 nm. The dispersions behave like Newtonian liquids up to 5 wt% silica content. At a higher silica content, depending on the shear rate range, they demonstrate a combination of non-Newtonian and dilatant behavior. Even higher silica content dispersions have a clearly defined yield point. The rheological behavior and morphology of silica dispersions in PAN solutions are primarily determined by the compound preparation method, and the search for the optimal procedure was the significant subject of the work. The quality of the composition obtained in each method was determined by a combination of rheological measurements and microscopy of samples, which renders it possible to develop an optimal method for silica introduction into PAN solutions. The relaxation times calculated for formulations with increasing silica concentration allow us to determine the concentration of percolation threshold (~3 wt%), which was confirmed by the viscosity increase and morphology of the film cast from the solution.

Coagulation simulation on a drop of silica-filled PAN solutions showed that the presence of silica particles lowers the appearance of defects during coagulation that may be useful in the wet spinning process.

The fibers were spun from concentrated PAN solutions containing silica, using wet and mechanotropic methods. It was found that silica contributes to a slight increase in the limiting draw ratio, while not affecting the elongation at break of the fibers. The mechanotropic method allowed us to obtain fibers with twice the strength compared with those obtained via wet spinning. The neat mechanotropic-made PAN fibers have a strength of ~800 MPa, while the fibers spun by the wet method only ~400 MPa. This could be explained by a much greater cross-section heterogeneity of the fibers and a higher degree of orientation drawing during as-spun fiber formation from the solution jet. The silica-filled fibers obtained by the mechanotropic method have high values of strength (~600 MPa), elastic modulus (~6 GPa), and elongation at break of about 25 wt%. It has been shown by TEM that silica in the fiber exists in the form of aggregates with dimensions of the order 50–150 nm that are uniformly distributed into fiber volume.

The results obtained open the possibility of choosing technological regimes for obtaining high-quality PAN fibers and films containing silica.

## Figures and Tables

**Figure 1 polymers-14-04548-f001:**
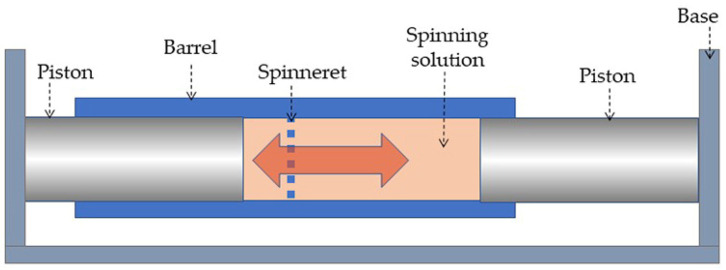
Scheme of the laboratory opposing mixer.

**Figure 2 polymers-14-04548-f002:**
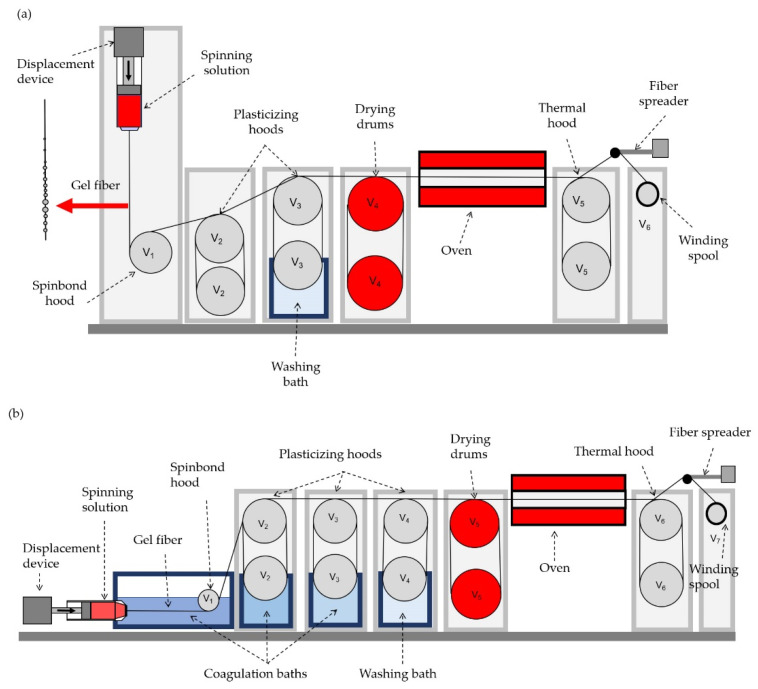
Schemes of the (**a**) mechanotropic and (**b**) wet spinning lines.

**Figure 3 polymers-14-04548-f003:**
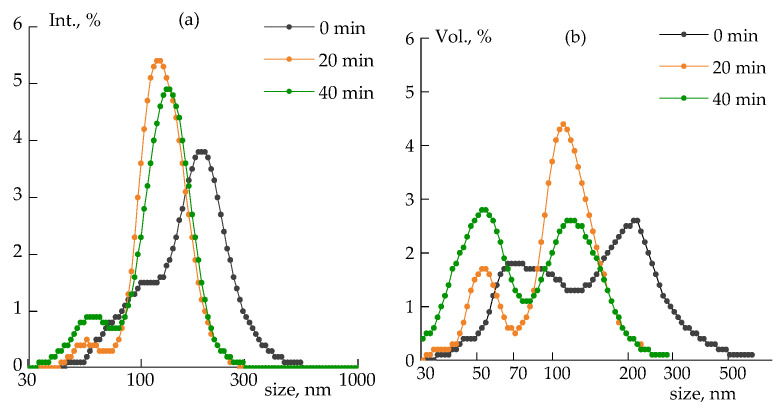
Size distribution of silica aggregates in DMSO in terms of scattered light intensity (**a**), and volume fraction (**b**) at the different sonication duration (indicated in plots).

**Figure 4 polymers-14-04548-f004:**
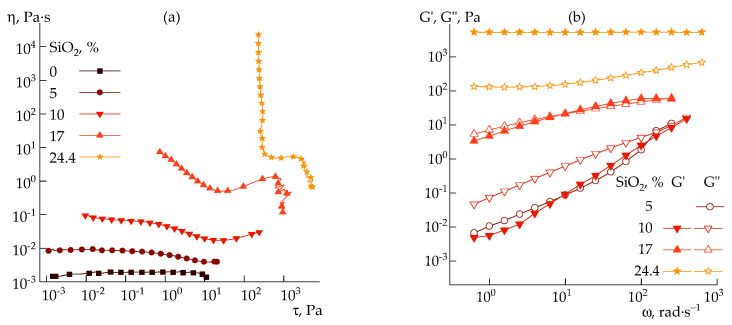
Flow curves (**a**) and frequency dependences of storage and loss moduli (**b**) for silica–DMSO suspensions of different concentrations (indicated in plots).

**Figure 5 polymers-14-04548-f005:**
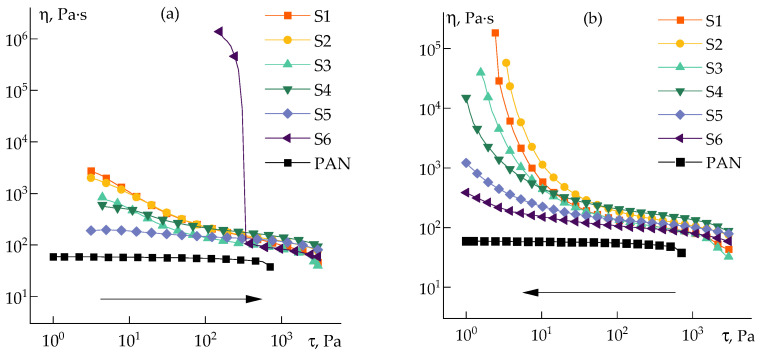
Flow curves of PAN solutions in DMSO containing 3 wt% of silica (at the solution weight) prepared by methods S1−S6. Experimental procedures: (**a**) at an increase in shear rate, (**b**) at a decrease in shear rate (starting from 1000 s^−1^).

**Figure 6 polymers-14-04548-f006:**
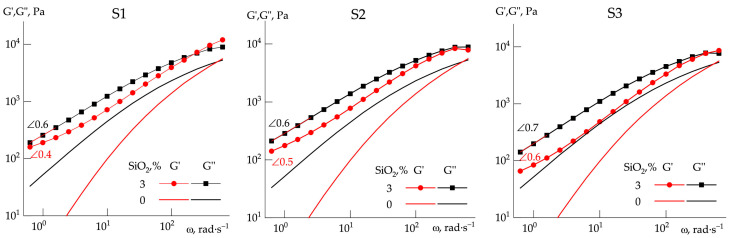
Frequency dependences of elastic and loss moduli for systems prepared by methods S1−S6. The slopes for the terminal regions are shown by “∠”. Silica concentration is at the solution weight.

**Figure 7 polymers-14-04548-f007:**
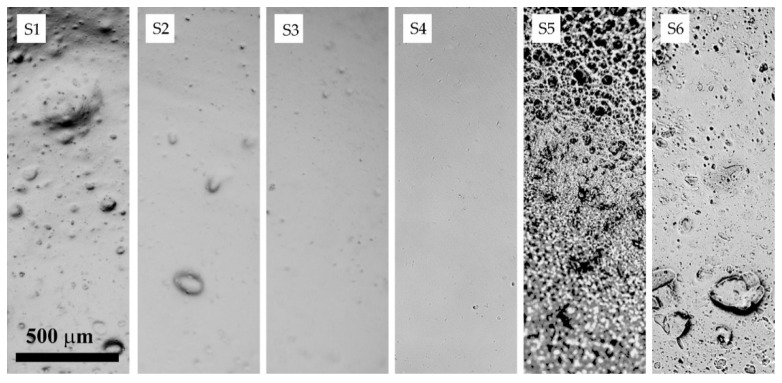
The morphology of the silica-filled PAN films obtained from systems S1 to S6. Images were made by optical microscopy.

**Figure 8 polymers-14-04548-f008:**
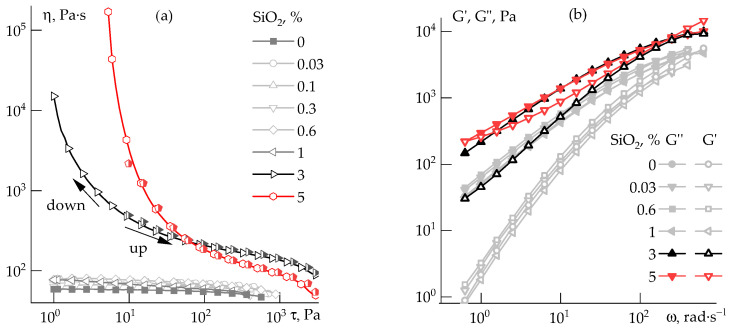
Flow curves at shear rate increase to 1000 s^−1^ followed by decrease in it (**a**), and frequency dependences of elastic and loss moduli (**b**) for 20 wt% PAN solutions with various silica content. Silica concentration is at the solution weight.

**Figure 9 polymers-14-04548-f009:**
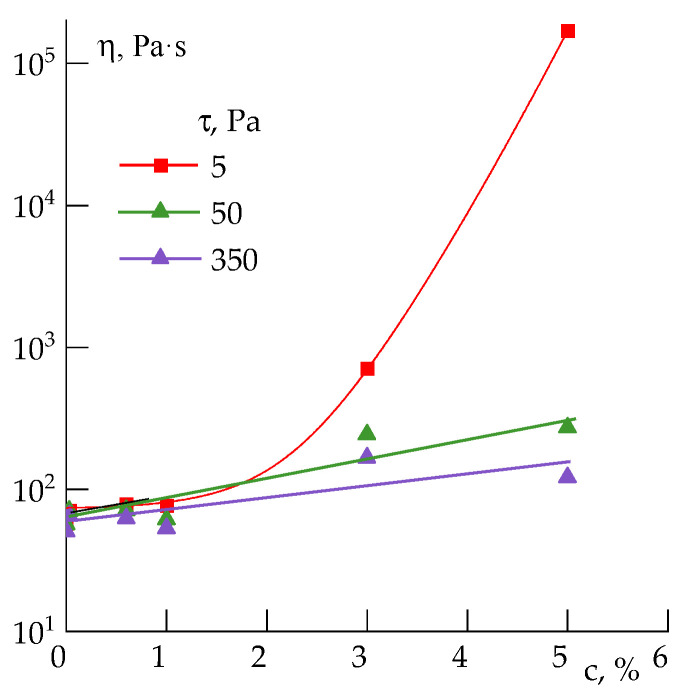
Dependence of viscosity on silica content in 20 wt% PAN solution at different shear stress. Silica concentration (c) is at the solution weight.

**Figure 10 polymers-14-04548-f010:**
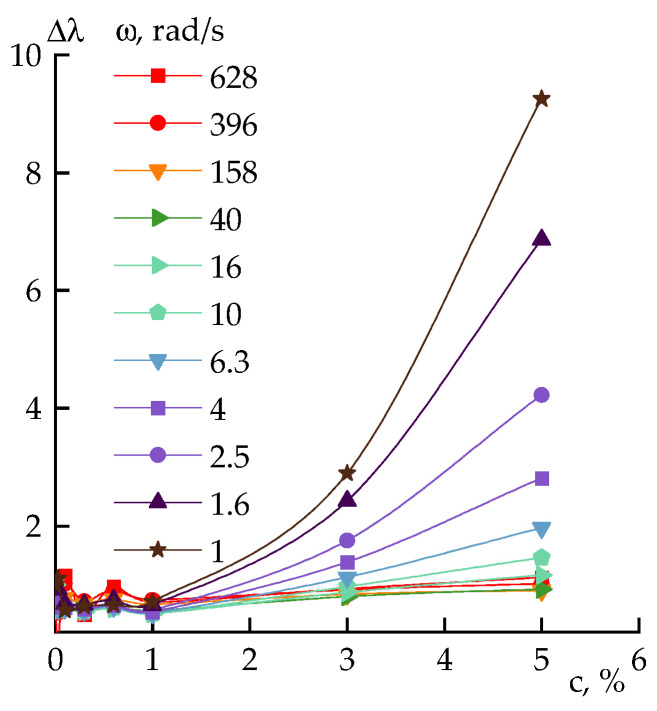
Dependence of the relative relaxation times on silica concentration at different frequency (indicated in plot). Silica concentration (c) is at the solution weight.

**Figure 11 polymers-14-04548-f011:**
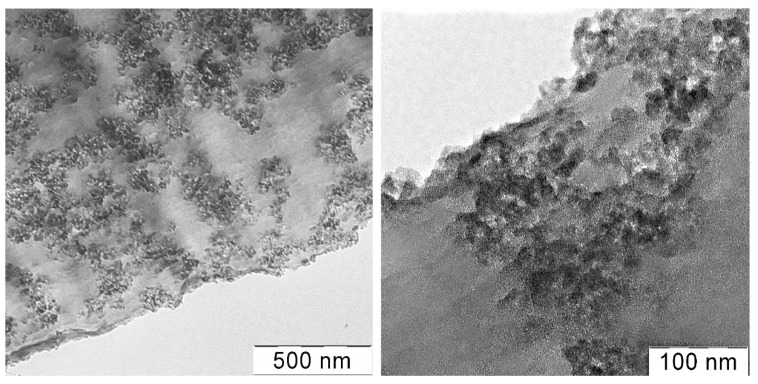
TEM images of a film section cast from a PAN solution containing 15 wt% silica (at the polymer weight) at various magnifications.

**Figure 12 polymers-14-04548-f012:**
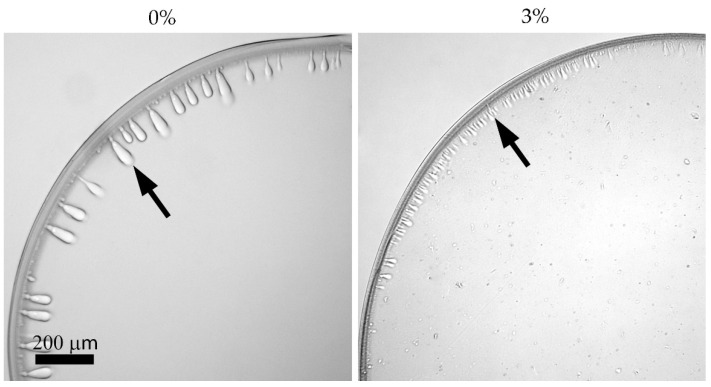
Optical microscopy images of droplets after the coagulation of a 20 wt% PAN solution with and without silica. Silica concentration is at the solution weight.

**Figure 13 polymers-14-04548-f013:**
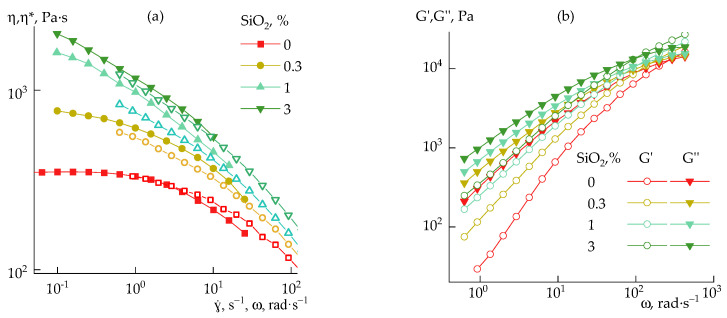
Flow curves measured with a stepwise increase in shear rate (shear viscosity is shown as open symbols and complex viscosity as filled symbols) (**a**) and frequency dependences of storage and loss moduli (**b**) of 33 wt% PAN solutions in DMSO with different silica content (at the solution weight indicated in graphs).

**Figure 14 polymers-14-04548-f014:**
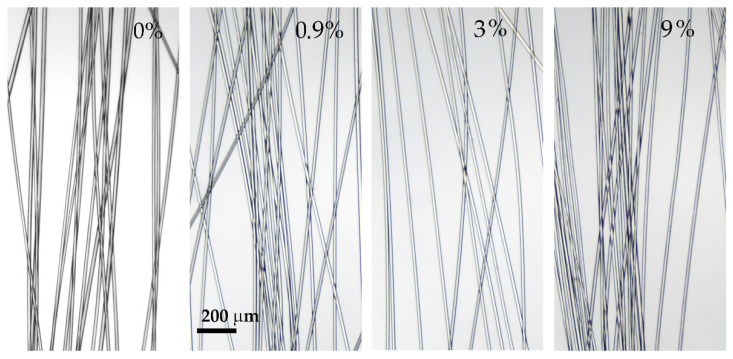
Images of spun fibers made by optical microscopy (at the polymer weight).

**Figure 15 polymers-14-04548-f015:**
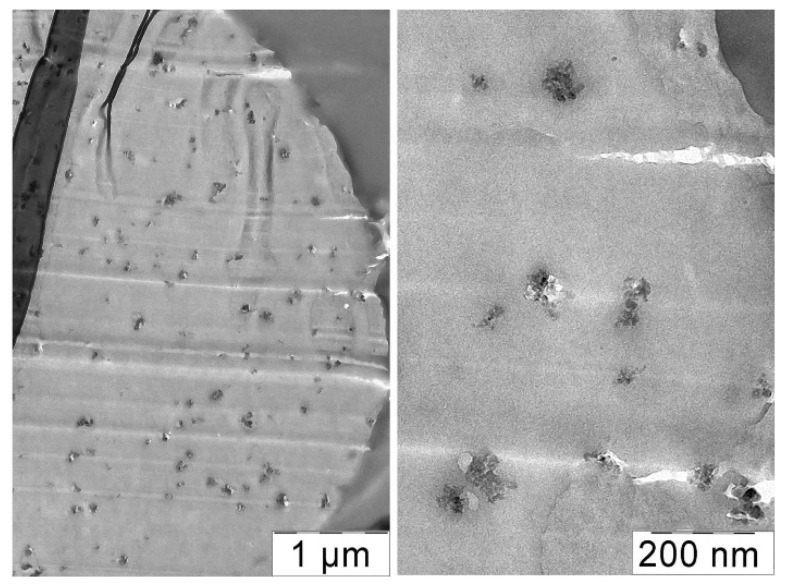
TEM images of PAN fiber cross-section containing 3% silica (at the polymer weight).

**Figure 16 polymers-14-04548-f016:**
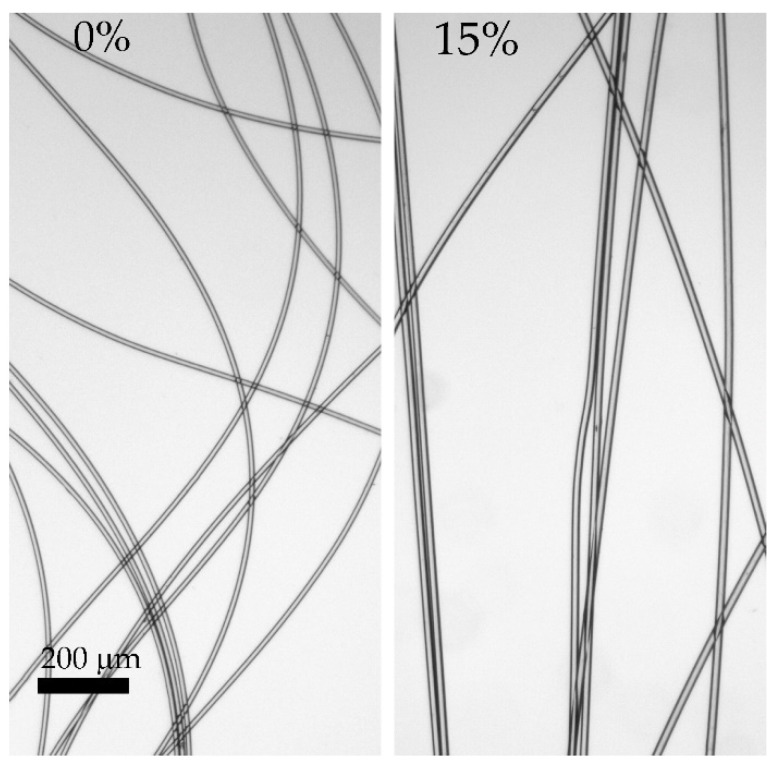
Images of spun fibers made by optical microscopy. Silica concentration is at the polymer weight.

**Table 1 polymers-14-04548-t001:** Scheme for the preparation of silica-filled PAN solutions. Plus means a particular treatment or ingredient was implemented, while minus means that it was not.

Sample:			S1	S2	S3	S4	S5	S6
Mixing sequence	1	+ Silica	+	+	+	+	+	+
	2	(a) ^1^ +PAN	−	−	−	−	+	−
	3	(a) +DMSO	+	+	+	+	+	+
	4	(b) Ultra-Turrax	−	−	+	+	−	+
	5	(c) Ultrasonic disperser	−	+	−	+	−	+
	6	(d) J-rotor, +DMSO	+PAN	+PAN	+PAN	+PAN	−	+33% PAN solution

^1^ (a) Mechanical mixing with a paddle mixer (Heidolph, Schwabach, Germany) for one hour at a speed of 1000 rpm; (b) homogenization in the operating unit of the “rotor-stator” mixer (ULTRA-TURRAX IKA T10 basic, Staufen, Germany) at 30,000 rpm with simultaneous exposure to ultrasound in the ultrasonic bath of Elmasonic S 15 (Elma Schmidbauer GmbH, Singen, Germany); (c) dispersion with a probe ultrasonic disperser UZDN-A (Lviv, USSR) with an exposure frequency of 22 kHz and a power of 130 W for 40 min; (d) mixing the slurry with the polymer using an anchor J-rotor with a rotor speed of 60 rpm for 24 h at 70 °C.

**Table 2 polymers-14-04548-t002:** Slopes’ values for 20 wt% PAN solutions filled silica. Silica concentration is at the solution weight.

SiO_2_, %wt	tan(G″)	tan(G′)
0−1	1 (±0.01)	1.8 (±0.01)
3	0.8 (±0.03)	1 (±0.05)
5	0.6 (±0.03)	0.4 (±0.05)

**Table 3 polymers-14-04548-t003:** Mechanical properties of fibers obtained from PAN solutions with silica by Mechanotropic spinning (at the polymer weight).

Sample, wt%	Diameter, µm	Tensile Strength, MPa	Elongation at Break, %	Modulus of Elasticity, GPa	Drawing Ratio	
					V_7_/V_1_	V_7_/V_spinneret_
0	13 (±3)	800 (±50)	18 (±3)	6 (±1)	6	330
0.9	14 (±1)	600 (±60)	21 (±3)	5 (±0.5)	6	330
3	11 (±0.3)	500 (±15)	23 (±1)	4.5 (±0.3)	7	420
9	15 (±1)	600 (±50)	21 (±3)	6 (±0.8)	7	420

**Table 4 polymers-14-04548-t004:** Mechanical properties of fibers obtained from PAN solutions with silica by wet spinning (at the polymer weight).

Sample, wt%	Diameter, µm	Tensile Strength, MPa	Elongation at Break, %	Modulus of Elasticity, GPa	Drawing Ratio	
					V_6_/V_1_	V_6_/V_spinneret_
0	12 (±0.5)	380 (±20)	19 (±2)	7 (±0.2)	6	70
15	19 (±1)	350 (±30)	20 (±3)	6 (±1)	6	70

## Data Availability

The data that support the findings of this study are available from the corresponding author upon reasonable request.

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
