# Peer review of "Silica-Filled Polyacrylonitrile Solutions: Rheology, Morphology, Coagulation, and Fiber Spinning"

_polymers, 2022, doi:10.3390/polym14214548_

Round 1
Reviewer 1 Report
In this manuscript, the authors studied the influence of the fumed silica on the morphology, coagulation processes, and rheological properties of suspensions in dimethyl sulfoxide (DMSO) and polyacrylonitrile (PAN)-DMSO solutions to produce composite films and fibers. They showed that silica DMSO concentrated suspensions (24 wt.%) form a weak gel with a yield point of about 200 Pa. At concentrations of ~5 wt.% and above the dispersions, depending on the shear stress, are pseudoplastic or dilatant liquids. A thixotropy appearance and a sharp increase in the relaxation time were observed for PAN solutions at a SiO2 content of more than 3−5%, which indicates the formation of structures with a gel-like rheological behavior. They investigated two spinning methods for preparing fibers: standard wet and mechanotropic. The results obtained open the possibility of choosing technological regimes for obtaining high-quality PAN fibers and films containing silica. This study is good and important to provide a promising strategy for producing composites of films or fibers. The interpretations of the results are well discussed. The quantity and quality of the figures are appropriate. We believe that this research subject is promising for developing composite films or fibers with controlling properties and functions.
Summary: I recommend publishing this manuscript after considering my comments on the attached file.

Author Response
Thank you very much for taking time to review our article. After correcting according to your remarks, it became better.
Line 413 What about the free volume size in your samples?
We did not really determine or calculate the free volume in the samples, focusing attention on the percolation threshold, which determines the rheological behavior of dispersions with clustered aggregates. Stable spinning of fibers from highly filled dispersions of PAN solutions is possible only at absence of aggregates. If you mean free volume in the fibers, sooner we need to measure the porosity. Usually, this value does not exceed 3-4%.
Line 422 The unit of the concentration, sometimes %, sometimes wt%, and sometimes %wt? All are similar?
All mass concentrations are now marked in the article so that there is no confusion.

Reviewer 2 Report
The manuscript entitled “Silica-filled Polyacrylonitrile Solutions: Rheology, Morphology, Coagulation, and Fiber Spinning” described an investigation of the influence of fumed silica on the morphology, coagulation behavior, and rheological properties of suspensions in dimethyl sulfoxide (DMSO) and polyacrylonitrile (PAN)-DMSO during the production of composite films and fibers. In particular, the rheological properties of silica-DMSO dispersions were investigated at various concentrations, and the dimensions of the silica aggregates were evaluated. In addition, the effect of silica presence on PAN solutions in DMSO were investigated with regard to the rheological properties, morphology, and interaction with a coagulant was investigated. Also, the appropriate spinning method for preparing composite fibers was investigated and selected, based on study of the structure, morphology, and mechanical properties of the thus-produced fibers.
Overall the research was well-executed, meaningful, and the research area is highly relevant to the field of Polymer Science as well as Materials Science. I believe that this work is suitable for publication pending minor revisions, as are suggested below.
Line 14: the phrase “that silica introducing. method into a PAN solution” is unclear.
Line 20: “By mechanotropic method” can be changed to “By the mechanotropic method” or possibly “With the mechanotropic method”.
Line 33: “To best accordance with the” can possibly be changed to “To best fulfil the” or “To achieve best accordance with the”.
Lines 50-51: A reference may be needed for the description of silica.
Line 66: “by the hydrogen bonds formation” can possibly be changed to “via the formation of hydrogen bonds”.
Line 76: “in that dispersions” can be changed to “in these dispersions”.
Line 91: “surface, attaching” can possibly be changed to “surface, such as by attaching”.
Line 100: “are promising for the” can be changed to “are promising candidates for the”.
Line 104: “may be contributed to the” can possibly be changed to “may be attributed to the”.
Line 153: The phrase “was diluted to 0.05 % silica for particle size measurements” may be a little unclear.
Line 157: “and PAN addition procedure into the blend” can possibly be changed to “and different procedures were also employed to added PAN into the blend”.
Lines 166-167, Table 1: Clarification may be needed with regard the the meaning of the plus and minus signs in Table 1, possibly as part of the notes below the table (I assume that plus means a particular treatment or ingredient was implemented, while minus means that it was not?).
Lines 231-232: “In the first one so-called by mechanotropic method, elaborated earlier in our laboratory” can possibly be changed to “The first method was the so-called mechanotropic method, which had been developed earlier in our laboratory”.
Line 248: “measured using Instron” can be changed to “measured using an Instron”.
Line 313: “solutions influences strongly their” can possibly be changed to “has a strong influence on their”.
Line 325: The phrase “liquids appearing the yield behavior” seems to be unclear.
Lines 341-342, Figure 6: Error bars may be needed for the data points in the plots shown in Figure 6.
Lines 374-375: The phrase “One of the limiting this issue is the rheological behavior” seems to be unclear.
Lines 377-378, Figure 8: Error bars may be needed for the data points in Figure 8.
Line 385: “easily broken at increase of” can possibly be changed to “easily broken with an increase of”.
Lines 391-393, Table 2: Error margins may be needed for the numerical values presented in Table 2.
Lines 399-400, Figure 9: Error margins may be needed for the data points in Figure 9.
Lines 403-404: “it has not been discussed the nature of the interaction between PAN solution and silica, important” can possibly be changed to “the nature of the interaction between PAN solution and silica has not been discussed, although it is important”.
Lines 423-424, Figure 10: error bars may be needed for the data points in Figure 10.
Line 439, Figure 11 caption: “Images of” can be changed to “TEM images of”.
Lines 455-456, Figure 12 caption: It may be necessary to clarify what type of microscopy was used to obtain these images. (I assume this was optical microscopy?)
Line 466: “at spinning” can possibly be changed to “during spinning”.
Lines 483-484, Figure 13: Error bars may be needed for the data points in Figure 13.
Line 537: “allows obtaining defect-free fibers, however, the maximum” can possibly be changed to “allows the preparation of defect-free fibers. However, the maximum”.
Line 538: “mechanotropic one” can be changed to “mechanotropic spinning”.
Line 551: The phrase “More high-filled systems” seems to be unclear, I am not sure what is meant here?
Line 567: “us to obtain twice more strong fibers compared with wet” can possibly be changed to “us to fibers with twice the strength compared with those obtained via wet”.
Author Response
On behalf of the entire team of authors, I would like to express my deep gratitude for the detailed review of the article. Now it looks much clearer and better for the readers.
Line 14: the phrase “that silica introducing method into a PAN solution” is unclear.
The phrase has been rewritten: It has been found that silica addition method into a PAN solution has a significant impact on the aggregates dispersibility
Line 20: “By mechanotropic method” can be changed to “By the mechanotropic method” or possibly “With the mechanotropic method”.
Thank you, the phrase is changed.
Line 33: “To best accordance with the” can possibly be changed to “To best fulfil the” or “To achieve best accordance with the”.
Thank you, the phrase is changed.
Lines 50-51: A reference may be needed for the description of silica.
The reference is added.
Line 66: “by the hydrogen bonds formation” can possibly be changed to “via the formation of hydrogen bonds”.
Thank you, the phrase is changed.
Line 76: “in that dispersions” can be changed to “in these dispersions”.
Thank you, the phrase is changed.
Line 91: “surface, attaching” can possibly be changed to “surface, such as by attaching”.
Thank you, the phrase is changed.
Line 100: “are promising for the” can be changed to “are promising candidates for the”.
Thank you, the phrase is changed.
Line 104: “may be contributed to the” can possibly be changed to “may be attributed to the”.
Thank you, the phrase is changed.
Line 153: The phrase “was diluted to 0.05 % silica for particle size measurements” may be a little unclear.
The phrase is rewritten.
Line 157: “and PAN addition procedure into the blend” can possibly be changed to “and different procedures were also employed to added PAN into the blend”.
Thank you, the phrase is changed.
Lines 166-167, Table 1: Clarification may be needed with regard the the meaning of the plus and minus signs in Table 1, possibly as part of the notes below the table (I assume that plus means a particular treatment or ingredient was implemented, while minus means that it was not?).
The information is added: Plus means a particular treatment or ingredient was implemented, while minus means that it was not.
Lines 231-232: “In the first one so-called by mechanotropic method, elaborated earlier in our laboratory” can possibly be changed to “The first method was the so-called mechanotropic method, which had been developed earlier in our laboratory”.
Thank you, the phrase is changed.
Line 248: “measured using Instron” can be changed to “measured using an Instron”.
Thank you, the phrase is changed.
Line 313: “solutions influences strongly their” can possibly be changed to “has a strong influence on their”.
Thank you, the phrase is changed.
Line 325: The phrase “liquids appearing the yield behavior” seems to be unclear.
Thank you, the phrase is changed.
Lines 341-342, Figure 6: Error bars may be needed for the data points in the plots shown in Figure 6.
Usually do not put error bars on frequency dependences, because these are very well reproducible data with errors of less than a few fraction of percent, imperceptible in logarithmic coordinates. Of course, the data are well reproduced only when systematic and instrumental errors are excluded and under conditions of homogeneity of the measured sample.
Lines 374-375: The phrase “One of the limiting this issue is the rheological behavior” seems to be unclear.
Thank you, the phrase is changed.
Lines 377-378, Figure 8: Error bars may be needed for the data points in Figure 8.
Usually do not put error bars on frequency dependences, because they are very well reproducible data with errors of less than a few fraction of percent, imperceptible in logarithmic coordinates. Of course, the data are well reproduced only when systematic and instrumental errors are excluded and under conditions of homogeneity of the measured sample.
Line 385: “easily broken at increase of” can possibly be changed to “easily broken with an increase of”.
Thank you, the phrase is changed.
Lines 391-393, Table 2: Error margins may be needed for the numerical values presented in Table 2.
The error margins are added
Lines 399-400, Figure 9: Error margins may be needed for the data points in Figure 9.
We checked that in this scale (4 orders of magnitude in viscosity), the measurement errors of the viscosity lie in an area slightly larger than the size of the dots, i.e. error bars are practically invisible on the graph. Some deviations in the viscosity of 3% samples at 50 and 350 Pa are probably due to some external factors which do not affect the general pattern of dispersion behavior.
Lines 403-404: “it has not been discussed the nature of the interaction between PAN solution and silica, important” can possibly be changed to “the nature of the interaction between PAN solution and silica has not been discussed, although it is important”.
Thank you, the phrase is changed.
Lines 423-424, Figure 10: error bars may be needed for the data points in Figure 10.
The data presented are based on well-reproducible data of frequency dependences of elastic and loss moduli. Adding error bars will of course somewhat improve the reliability, but it will significantly complicate the perception of the Figure. Moreover, it is not the absolute values of the specific relaxation times but their relative values to identify the observed qualitative transition at 3% and 5% of silica concentration.
Line 439, Figure 11 caption: “Images of” can be changed to “TEM images of”.
Thank you, the phrase is changed.
Lines 455-456, Figure 12 caption: It may be necessary to clarify what type of microscopy was used to obtain these images. (I assume this was optical microscopy?)
Yes, it is optical microscopy, the method is added.
Line 466: “at spinning” can possibly be changed to “during spinning”.
Thank you, the phrase is changed.
Lines 483-484, Figure 13: Error bars may be needed for the data points in Figure 13.
Usually do not put error bars on frequency dependences, because these are very well reproducible data with errors of less than a few percent, imperceptible in logarithmic coordinates. Of course, the data are well reproduced only when systematic and instrumental errors are excluded and under conditions of homogeneity of the measured sample.
Line 537: “allows obtaining defect-free fibers, however, the maximum” can possibly be changed to “allows the preparation of defect-free fibers. However, the maximum”.
Thank you, the phrase is changed.
Line 538: “mechanotropic one” can be changed to “mechanotropic spinning”.
Thank you, the phrase is changed.
Line 551: The phrase “More high-filled systems” seems to be unclear, I am not sure what is meant here?
This refers to dispersions with a higher silica content.
Line 567: “us to obtain twice more strong fibers compared with wet” can possibly be changed to “us to fibers with twice the strength compared with those obtained via wet”.
Thank you, the phrase is changed.
